# Chromosome Dynamics Regulating Genomic Dispersion and Alteration of Nucleolus Organizer Regions (NORs)

**DOI:** 10.3390/cells9040971

**Published:** 2020-04-15

**Authors:** Hirohisa Hirai

**Affiliations:** Primate Research Institute, Kyoto University, Inuyama, Aichi 484-8506, Japan; gibbonhiro13@gmail.com

**Keywords:** acrocentric chromosome association, subtelomeric association, chromosomal configuration, ectopic recombination, genomic dispersion, primates, bulldog ants

## Abstract

The nucleolus organizer regions (NORs) demonstrate differences in genomic dispersion and transcriptional activity among all organisms. I postulate that such differences stem from distinct genomic structures and their interactions from chromosome observations using fluorescence in situ hybridization and silver nitrate staining methods. Examples in primates and Australian bulldog ants indicate that chromosomal features indeed play a significant role in determining the properties of NORs. In primates, rDNA arrays that are located on the short arm of acrocentrics frequently form reciprocal associations (“affinity”), but they lack such associations (“non-affinity”) with other repeat arrays—a binary molecular effect. These “rules” of affinity vs. non-affinity are extrapolated from the chromosomal configurations of meiotic prophase. In bulldog ants, genomic dispersions of rDNA loci expand much more widely following an increase in the number of acrocentric chromosomes formed by centric fission. Affinity appears to be a significantly greater force: associations likely form among rDNA and heterochromatin arrays of acrocentrics—thus, more acrocentrics bring about more rDNA loci. The specific interactions among NOR-related genome structures remain unclear and require further investigation. Here, I propose that there are limited and non-limited genomic dispersion systems that result from genomic affinity rules, inducing specific chromosomal configurations that are related to NORs.

## 1. Introduction

Fontana described the nucleolus for the first time in 1781 and formally named by Valentine in 1836, and these early studies utilized germinal vesicles to observe and examine the nucleolus [1]. Later historical studies of the nucleolus organizer regions (NORs) from the 18th century to the beginning of the 20th include investigations of nuclear composition and relations to chromosomes [2]. Cytogenetic staining techniques were later used to detect the organelle and they focused on its nature and organization [3]. Newly established techniques, such as in situ hybridization, proved that NORs are conspicuous structures at the secondary constrictions of chromosomes, the actual sites of the ribosomal genes [2,4,5], and they contain the tandem rDNA arrays that are responsible for composing the nucleolus, which is the site of synthesis and processing of ribosomal proteins and rRNA [6]. Variations in the localization and transcriptional activity of NORs have been well documented in many species and/or phyletic groups. Location can be detected by in situ hybridization techniques using multicopy DNA clusters of rDNA (especially 18S and 28S) [7] and activity can be detected with silver nitrate (Ag-NOR) staining techniques, because NORs have an argyrophilic property of stainability to non-histone proteins in sulfhydryl and disulfide groups [8,9,10]. NORs of humans have been most extensively investigated with respect to locality and activity; the results indicate localization at five chromosome pairs (ten acrocentrics): numbers 13, 14, 15, 21, and 22 [4,11]. A statistical analysis of chromosome associations revealed that the NOR chromosomes were those most commonly involved in Robertsonian translocations, and that such breakages might be related to acrocentric associations [12,13]. The techniques of in situ hybridization and Ag-NOR staining have been improved over the years and, thus, the function and localization of NORs have become more precisely determined. Methods combining multiple techniques have provided greater detail on the physical and molecular features of NORs and proved that they have developed, and are maintained by, a system of concerted evolution [14,15,16,17,18]. The chromosome localization of rDNA using fluorescence in situ hybridization (FISH) or Ag-NORs has been investigated in many organisms. However, there remain some enigmatic phenomena, like specific NOR localities and/or their relationships with karyotypic changes. Thousands of cytological observations using these techniques suggest that there are some natural “rules” governing the localization and/or variability of NORs. Therefore, my colleagues and I have studied these characteristics of NORs in several phyletic groups. In this short paper, I will explain the underlying mechanisms of some chromosome dynamics—the interaction or non-interaction among chromosomes through meiotic chromosomal configurations—and infer genomic dispersion systems for rDNA and other repeat arrays comprising constitutive heterochromatin. My explanations are informed by our observations of primates, tree shrews, and ants. I propose systems of genomic dispersion of rDNA that are rooted in a concept of “affinity” and “non-affinity” among such repetitive arrays, being accompanied by “molecular” and “site” effects. “Affinity”, as used here, is defined as an attractive force, or junction power, among the DNA segments.

## 2. Methods Used for Inferring Genomic Dispersion Mechanism

Eukaryotic rDNA consists of tandem repeat gene segments that are composed of 18S, 5.8S, and 28S tracts. My colleagues and I therefore used probes of human 18S rDNA (kindly supplied from Dr. AK Godwin) and ant 28S rDNA clones (*M. closlandi* (2n = 2) rDNA (pMc.r2)), respectively, in FISH analysis to detect localization of NORs in mammals and ants [19,20]. Probes of other repeat arrays often found in a similar chromosome position as rDNA arrays (e.g., subterminal satellite, β-satellite) were also used to detect their respective locations, as well as their locations relative to rDNA arrays. FISH analysis is dependent on chromosome preparations. The samples obtained from cell culture (details provided in original papers) were comparatively stable, but chromosome samples that were obtained from in vivo tissues require careful treatments. My colleagues and I developed chromosome preparation methods for such in vivo samples (testis, ovary, and/or ganglion) obtained from small animals, like insects and parasites. The technique was originally developed for ant chromosomes [21], and I modified it in order to obtain chromosome preparations from a small biopsy of primate testis tissue that were collected under anesthesia [22,23]. Ag-NOR staining for detecting transcriptional activity of NORs has been gradually improved from early techniques [8,9]. My colleagues and I used a simple and quick technique modified from a gelatin/silver nitrate method [24], which has proven to be very effective in obtaining clear Ag signals.

## 3. Relationship of rDNA and Other Repeat Arrays

NORs sometimes exhibit different localization patterns, even among closely related taxa. For instance, humans and chimpanzees signal positive in five pairs (10 chromosomes) for rDNA probes, but gorillas are positive only in two pairs (four chromosomes) and orangutans in eight pairs (16 chromosomes) [16,19]. Though these are all ape species, their NOR patterns show either constancy (i.e., human & chimp) or non-constancy (the other apes). The cytological and genomic causes of these differences are not well understood. Therefore, I discuss here examples of NOR differences in some organisms and infer the mechanisms of their alteration. Indeed, the specific patterns of genomic dispersion of NORs must reflect the conditions of their preservation, as well as those that induce variability. Insight is drawn from previous and recent studies.

Recently, my collaborators and I encountered several examples of affinity and non-affinity between rDNA and other arrays located at chromosome ends. In one example, we observed the chromosomal configuration of subterminal satellites (StSat)—a repeat array that is specific to African apes—and inferred their evolutionary aspects together with those of rDNA arrays [25]. Other investigations have discovered the variability of rDNA localization and NOR loci in the great apes (orangutans, gorillas, chimpanzees, and bonobo), and humans [25,26,27]. Those investigations found that gorillas have only two pairs of loci with rDNA arrays, while the other hominoids have considerably more (eight or five pairs), as mentioned earlier [19,26]. Interestingly, our recent research [25] demonstrated that gorillas have StSat loci at one or both ends of all chromosomes, but none of the StSat loci colocalize with rDNA arrays in somatic metaphase chromosomes. This strongly suggests the following. Although StSat terminal blocks have firm associations with each other, as seen in the chromosomal bouquet at meiotic prophase (affinity structure), such StSat associations exclude rDNA arrays (non-affinity structure). Also of note, the short arms of the six gorilla acrocentrics without either StSat or rDNA loci harbor unknown repeat DNA. Thus, unknown repeat DNA and StSat loci are both exclusive to rDNA arrays. Figure 1 depicts chromosomal configurations postulated from observations of molecular cytogenetic studies.

Chromosomal configuration can be regarded as bouquet configuration that formed through telomere clustering and binding to the inner surface of the nuclear membrane [28,29]. This configuration is formed by the polymerization of telomere binding protein (e.g., TERB1) and accumulation of the transmembrane complex at the telomere association sites [30,31]. The protein molecule complex is important as a system that facilitates chromosome alignment to promote the pairing of homologous chromosomes and simultaneously agitates chromosomes to interrupt transient nonhomologous interaction, ectopic recombination between dispersed repetitive DNA [32,33]. It also acts as a central hub for the assembly of a conserved meiotic telomere complex that is required for chromosome movements [31]. In a representative model plant *Arabidopsis*, the bouquet configuration is functionally formed by nucleolus-associated telomere clustering. That is, the pairing of telomeres seems to be a homology-dependent process involving immediately sub-telomeric DNA sequences [34]. Accordingly, the clustering of chromosome terminal regions initiated by the meiotic proteins (the telomere “bouquet”) is an important process in meiosis for regulating chromosome movements and alterations, as well as interactions between homologous chromosomes or heterologous chromosomes. Sub-terminal DNA sequences probably also contribute to chromosomal configuration and interaction via specific molecular features, although more detailed studies are needed.

These observations strongly suggest that the physical relationships among rDNA and other repeat arrays can act as natural constraints to the genomic dispersion of some repeat segments. There seems to be a natural ‘law’ against association of segment blocks with different DNA motifs, even if they are both constitutive heterochromatin. In the gorilla example, not only does this species have merely two rDNA loci, it also has only one translocation [26]. The limited number of rDNA loci and translocations is likely a reflection of the limited association and recombination between rDNA and other repetitive segments in meiotic prophase. If follows that an increase in affinity naturally leads to a higher rate of recombination since the recombination rate likely depends on the degree of affinity among subterminal regions. This inference is reinforced by other evidence presented in the next sections.

## 4. Manifestations of Chromosomal Configuration in Meiotic Prophase

Figure 1 illustrates examples of variable chromosomal configurations in meiotic prophase that were observed in and inferred from previous studies. An association among rDNA arrays (*RA* + *RA*) is often observed in animals and it is likely the engine of concerted evolution of rDNA, leading to sequence homogenization by interchromosomal recombination. The association (*RA* + C*HA*) between *RA* and constitutive heterochromatin (containing unidentified) arrays (C*HA*) is occasionally observed in meiosis and it has a shape similar to *RA* + *RA*, but its function is thought to be that of an engine for genomic dispersion of rDNA through ectopic recombination. The rate of ectopic recombination is much less than that of homologous recombination, as inferred from successive observations indicating that heterozygotic variations of rDNA localization are sporadic, as mentioned above. Indeed, double-strand breaks (DSBs) at the sites of repetitive sequences can be a potent source of genomic alteration [35,36]. The local chromatin environment very likely plays a role in promoting the repair of DSBs within rDNA via the homologous recombination pathway [36].

In studies of the African apes, there was no association (*RA*//*StSat*) between *RA* and subterminal satellite repeats (*StSat*) in somatic metaphases and meiotic prophases of chimpanzees, and somatic metaphases of bonobo and gorilla. The rDNA and StSat arrays were also never found to colocalize in those studies. Similarly, there was no association (*RA*//*URA*), or colocalization, between *RA* and unknown repeat arrays (*URA*) in meiotic cell division. These different repeat arrays (e.g., *RA* and *StSat*) are mutually exclusive and likely do not come into contact during any stage of chromosome cell division. However, subterminal satellites may operate a bit differently than other repeats. They will not only associate with each other (*StSat* + *StSat*)*,* they will also form associations (*StSat* + *NRS*) with non-repeat segments (*NRS*) of different chromosomes, thereby laying the foundation for genomic dispersion and concerted evolution of StSat arrays, similar to rDNA arrays. Although unknown repeat arrays (*URAs*) that were observed as heterochromatic blocks were not identified at the sequence level in our previous studies, some satellite repeats with motifs, like (AATGG)_n_, (ACTCC)_n_, and (AAAG)_n_, are suggested as candidates based on recent genome sequence analysis [37].

Chromosomal configurations formed through clustering of repeat sequences in cell division provide insight into overall chromosome evolution [38]. The configurations that are illustrated in Figure 1 have been presented as evolutionary aspects of repeat DNA arrays in previous studies. Indeed, *RA* + *RA* and *RA* + *CHA* associations have been observed in many organisms and their roles have been described in the context of genome construction and evolution. For instance, the association and post-association dispersion of multisequence families have been noted and discussed in studies of somatic chromosomes in humans and apes [16,19,39,40,41,42,43,44]. The existence of non-association (exclusion) systems between specific repeat arrays has also recently been found in studies of meiotic cell division [25]. The *RA*//*StSat* exclusion and the *RA*//*URA* exclusion, both being demonstrated in Figure 1, are representatives of such systems. We observed these phenomena in the meiotic prophase of chimpanzees and inferred the same exclusions from localization patterns in somatic metaphase chromosomes.

By comparison, the Old World monkeys have interstitial (*INTRA*) rDNA arrays, and these can be located either in metacentric (m) chromosomes or acrocentric (a) chromosomes (*INTRA*m or *INTRA*a) (Figure 1). Interestingly, the monkey species examined to date do not show the genomic dispersion of rDNA arrays [45]. The interstitial rDNA regions that were observed in *INTRA*m and *INTRA*a hardly make association between each other because proximal regions of chromosome arms cannot easily make contact with other regions (unlike the telomere-telomere associations in terminal regions), though self-duplication at the locus seems to arise [46]. Taken together, these observations suggest that the prevention of association and post-association dispersion of rDNA arrays seem to be controlled by two mechanisms: a ‘molecular effect’ (avoidance by molecular property) and a ‘site effect’ (avoidance by locational property). Thus, the chromosomal location of an rDNA array might preclude its movement elsewhere in the genome, even if karyotype change occurs in an organism (Figure 2a).

## 5. Conditions of the “Molecular Effect” System of Genomic Dispersion

The affinity of particular genetic blocks, e.g., *RA* + *CHA*, *RA* + *RA*, *StSat* + *NRS*, and *StSat* + *StSat*, facilitates genomic dispersion and concerted evolution. Thus, it plays an important role in overall genome construction and it places limitations on dispersion inherent to each type of repeat block. However, we encountered a special case that appears to represent a “no-limit” system of genomic dispersion: the rDNA localization of Australian bulldog ants (genus *Myrmecia*). This ant species complex is very useful for investigating chromosome evolution, because they are a species complex with similar morphology, but a wide range of chromosome numbers (2n = 2 − 76) [48]. This is a prominent example that demonstrates the relationship between rDNA genomic dispersion and karyotypic evolution [47]. In our previous works, we examined 16 *Myrmecia* species consisting of three species groups: *M. gulosa* (six species), *M. mandibularis* (two species), and *M. pilosula* (eight species). In this case, a gain of chromosome number due to centric fission proportionally produces many more rDNA loci (Figure 3). The increase of rDNA loci must be a result of terminal association and subsequent recombination between rDNA and constitutive heterochromatin arrays (Figure 1, *RA* + *CHA*). This is a case of a positive molecular effect for genomic dispersion: more acrocentrics lead to more rDNA loci. The reason why is that the number of rDNA-chromosomes is two in the “primitive” karyotypes 2n = 2 − 8, but it has gradually increased from two to 19 in karyotypes with 2n = 10 − 32 and 2n = 38 − 76. Figure 3 depicts the increase of diploid number (two to 76)—via an increasing number of acrocentrics by centric fission—as correlated with change of rDNA-chromosome number (two to 19) [47]. However, if the heterochromatin arrays of the acrocentric short arms did not have affinity with the rDNA arrays, the rDNA arrays would not have been able to move to those short arms via association and ectopic recombination. Thus, the molecular effect on genomic dispersion of rDNA arrays seems to be regulated by the structure of heterochromatin or possibly by unknown repeat arrays that were acquired in the short arm of acrocentrics formed by centric fission. The postulated pathway of genomic dispersion is depicted along with karyotypic changes in Figure 2b.

The regulation of genomic dispersion by the molecular effect is also implicated in studies of rDNA localization in orangutans, which have eight pairs of NORs when compared to the other great apes, which have locations on only five (chimpanzees & bonobos) and two (gorilla) chromosome pairs [19]. Interestingly, orangutans (Asian great apes) have no StSat blocks, whereas these arrays are extensive in all of the African great apes [25,49]. This situation suggests that the genomic dispersion of rDNA is limited in African apes by molecular effect: in this case, non-affinity with StSat. Orangutans, by comparison, do not have such strong limitation on dispersion of rDNA for three possible reasons: absence of StSat arrays, preservation of rDNA, and/or acquisition of affinity association. Yet, non-affinity conditions among the array elements of some chromosomes may still provide indirect insight. With these conditions in effect, for example, the origin of rDNA on the long arm end of the (Sumatran) orangutan Y-chromosome appears to be associated with a translocation event [50]. Such phenomena strongly suggest that the molecular effect results from the affinity and non-affinity associations between rDNA and other repeat arrays and can, thus, explain the presence of (many vs. few) rDNA arrays.

However, if this understanding of the limitations to genome dispersion of NOR loci is correct, the localization patterns of human rDNA require more explanation. Humans do not have StSat loci, but their rDNA arrays are restricted to the short arms of the five acrocentrics; they have not dispersed elsewhere. The simplest possible explanation is existence of other arrays that share the same non-affinity with rDNA as StSat blocks—a non-affinity system that is different from those of the African great apes. When considering that humans have non-heterochromatic DNA arrays, such as ß-satellite DNA and other repetitive sequences at the distal part of NORs [25,51,52], the distal structure might have a property that makes such regions incompatible with the terminals of other chromosomes, forming a non-affinity system. This notion is corrobortated by differences in the NOR activity of rDNA regions between humans and chimpanzees. Namely, the loss of rDNA, which is one of the means of inactivation of NORs, occurs approximately seven times more frequently in chimpanzees than in humans [44]. This suggests that chromosomal regions carrying NORs in chimpanzees likely interact more frequently with other chromosomal ends compared to similar regions in humans. Thus, chimpanzees more easily experience loss or gain of rDNA than humans. However, previous observations suggest that the presence of StSat likely prevents unchecked (“no-limit”) genomic dispersion of rDNA in chimpanzees. Thus, humans and chimpanzees very likely regulate the dispersion of rDNA differently due to differences in genome composition. This inference is supported by a recent study of fine genome structure around NORs in humans and chimpanzees [53]. In this study by van Sluis et al., a distal junction (DJ) DNA sequence block was assembled as a contig sequence block in human acrocentrics that abuts rDNA arrays on their telomeric side. Chimpanzee acrocentrics, however, possess a DJ-like block abutting rDNA arrays that are inverted as a unit compared to human acrocentrics. In another study, proximal junction (PJ) sequence blocks abutting rDNA arrays adjacent to centromeres were also observed by genome assembling [54]. Indeed, human and chimpanzee acrocentrics, respectively, have an ordering of centromere-PJ-**rDNA-DJ**-telomere and of centromere-PJ-**DJ-rDNA**-telomere, which is in accordance with our previous observations. Such differentiations of genome structures may introduce structural and functional variations between chromosomes and among individuals.

## 6. Variation in rDNA Loci

The terminal regions usually undergo telomere-telomere clustering during early prophase of the first meiotic cell division, forming the so-called meiotic bouquet [55]. The bouquet probably is the starting point of association and genomic dispersion for rDNA and other repeat arrays—actions that are initiated by telomere-telomere clustering. Association is a trigger for making homologous and ectopic pairing (and synapsis) much faster and more efficient [56]. The strong affinity of rDNA arrays is likely the cause of random associations between acrocentric chromosomes—predicted by mathematical theory [57,58]—which, in turn, sets the stage for the most frequent chromosomal rearrangement in humans, Robertsonian translocations. The association of different chromosomes leads to increased variability around NORs, and the proximity of the rDNA arrays and then predisposes them to recombination at the associated terminal regions. If there is an error in the ‘molecular effect’ at terminal associations (i.e., breakdown of the system of avoidance among distinct genome components), recombination will bring about new intra-specific variations that result from such atypical relationships among rDNA and other repeat arrays [44,59].

Usually, rDNA locations are found to be highly conserved and/or have readily traceable divergence patterns among closely-related species [4,19,60,61]. However, there are some cases in which very different rDNA patterns—without readily traceable intermediate stages—have been observed. In two sibling species of tree shrew, for example, one form (*Tupaia belangeri*) only has four rDNA loci, whereas another closely-related taxon (*T. glis*) has between seven and eight such loci [62]. Likewise, Indian pygmy field mice show an unusually high number (9–34) of NORs throughout their species complex [63]. The cytogenetic differences cannot easily be explained by an affinity system in either example. In these cases, where molecular arrangements in NOR arms have outpaced morphological differentiation, it is likely that some other cytogenetic mechanism is, in effect, such as transposition of NORs. Indeed, transposition with mobile elements, rather than structural rearrangement, is known to be a decisive mechanism for rDNA genomic dispersion in plants [64,65,66,67,68], insects [69,70,71], and mammals [63]—and it might occasionally operate as a mechanism of the “molecular effect” in organisms. Certainly, changes of the rDNA sites do not vary randomly and seem to be mainly mediated by the bouquet configuration because the sites are preferentially localized at the terminal regions (73.8%) [72]. Yet, since the mammals surveyed to date do not exhibit direct association between transposable elements and rDNA arrays [73], it is most likely that the affinity/non-affinity system described here between rDNA and other repeat arrays is driving related cytogenetic changes. This system probably prevents excessive change of NORs in mammals, and instead regulates the comparatively steady genomic dispersion of rDNA arrays. However, more studies are needed to provide a robust test.

Intra-specific variations of rDNA loci also appear to occur by the ‘site effect’. Observations with FISH and Ag-NOR methods in humans and chimpanzees suggest that structural changes of terminal regions induce intraspecific differences in transcriptional activity (expression or repression) of rDNA. Analyses of 48 humans and 46 chimpanzees demonstrated that 54.2% of the former and 10.9% of the latter lacked an rDNA locus on one or more acrocentrics [44]. Although these closely related species both generally show 10 chromosomes (five pairs) to be positive for rDNA loci, each nonetheless exhibits unique patterns of intraspecific variation. Molecular cytogenetic analyses revealed that the differences between the two species in the terminal NOR regions of acrocentric chromosomes stem from structural differences. The regions just inside of the telomeres of the short arms in these chromosomes consist of rDNA arrays in chimpanzees (rDNA + rDNA), but other repeat arrays in humans (ß-satellite + rDNA) [19,44]. Critically, these differences exist in the chiasma-free zones (CFZ), the terminal 0.5–1.0% of each chromosome, in which no point of crossover contact occurs [74]. In other words, because chimpanzees retain rDNA arrays within the CFZ, the arrays will be maintained in their original place, even after recombination. By comparison, because humans retain rDNA arrays proximal to (outside of) the CFZ, the arrays will occasionally be recombined and transferred to other regions by cross over, leaving other repeat arrays in their place. Thus, in subsequent generations, chimpanzees persistently retain the rDNA arrays in this terminal region (CFZ), whereas humans retain the other repeat arrays (see Figure 4). In addition to the physical differentiation of rDNA loci described, both species exhibit polymorphisms that are related to transcriptional inactivation of NORs. An earlier analysis suggested that transcriptional inactivation in humans resulted from the loss of the rDNA loci, whereas the inactivation in chimpanzees likely resulted from the repression and/or methylation of rDNA [44]. The methylation system that we detected on chimpanzee chromosomes was suggested as a mechanism to the inactivation of rDNA genes and a means of propagating transcriptional silencing in the mouse [75].

## 7. Conclusions

Here, I present a working theory of “affinity/non-affinity” and “molecular/site effect” systems that govern the genome dispersion of rDNA and other repeat arrays in animals, using primates and insects as examples. Although studies of rDNA dispersion mechanisms are still limited in number, the explanations that are provided here are informed by chromosomal configurations and behaviors directly observed in meiotic cell division and inferred from the genomic organization of chromosome structures examined in FISH analyses (Figure 1). These findings suggest that rDNA location patterns evolved via genomic dispersion attended by ectopic association, subsequent recombination among rDNA and other repeat arrays, and continual modification by concerted evolution. The genomic dispersion of rDNA arrays has been maintained in many animal taxa and it is likely to operate in conjunction with karyotype evolution, as depicted in Figure 2. I inferred that conditions of genomic affinity vs. non-affinity regulate localization and dispersion of NORs. Affinity among NORs and other repeat arrays induces association and recombination in meiosis, which, in turn, lays the foundation for genomic dispersion, followed by concerted evolution and genomic homogenization. Non-affinity among NORs and other repeat arrays is a condition in which the lack of contact with arrays on different chromosomes consequently limits genomic dispersion, thus resulting in specific localization of NORs.

Affinity and non-affinity systems are pivotal mechanisms of “molecular effect” for genomic dispersion in repeat arrays. The “site effect” is a precondition of genomic dispersion and non-dispersion; specifically, chromosome terminal sites can induce associations by frequent contact according to characteristic chromosome behavior in meiotic cell division, but proximal internal sites of chromosomes cannot induce association, because they have little chance of contact in meiosis. Thus, chromosomal meiotic configuration must be carefully and critically analyzed. In addition to primates, other animal models are needed to test and expand our understanding of genome dispersion and overall chromosomal meiotic configuration. The development of such comparative animal models first requires the identification of repeat arrays in those species. Further, both intensive and extensive investigations of genomic structure around NORs, and meiotic chromosome behavior in prophase, are needed in order to develop a more thorough understanding of the affinity vs. non-affinity systems for NOR genomic dispersion and variability of transcriptional activity.

Physical contacts between chromatins are related to their tropism by the affinity/non-affinity system, which stems from DNA traits. This ultimately brings about genomic reaction, including dispersion and/or recombination, and generates genetic divergence. Structural patterns also likely represent important indexes that reflect the relationship between transcriptional activity and NOR evolution. The most direct connection is that between NORs and nucleolus. In cytological observations of Australian bulldog ants, the relationship between nucleolus formation and rDNA loci revealed that the nucleolus (visualized by silver nitrate staining) only appeared in the vicinity of an rDNA locus in interphase to prophase. That is, only an rDNA array associates with the active nucleolus in the formation of nucleolus. This phenomenon was observed in all examined species [20]. The localization of NORs in the nucleolus, and its associations with the nucleolus, are closely linked to the transcription activity of rDNA [76]. Indeed, future chromosomal studies that are related to rDNA and NORs need to pay close attention to cellular states and epigenetic states of the genome that are driven by the rDNA/nucleolus [77].

## Figures and Tables

**Figure 1 cells-09-00971-f001:**
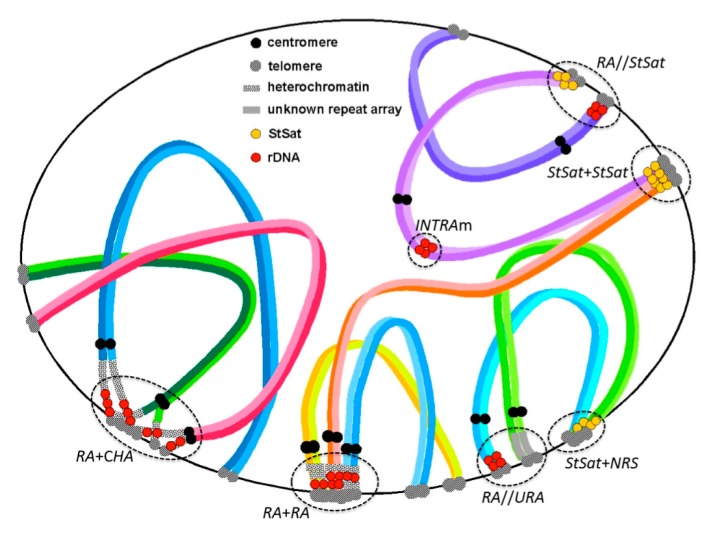
Examples of chromosomal configuration in pachytene stage inferred from observations of somatic and meiotic chromosomes in primates. *RA* + *RA*: association among rDNA arrays (*RA*) (observed in chimpanzee). *RA* + C*HA*: association between *RA* and constitutive heterochromatin arrays (C*HA*) (observed in slow loris). *RA*//*StSat*: non-association between *RA* and subterminal satellite (StSat) (observed in chimpanzee, bonobo, and gorilla). *RA*//*URA*: non-association between RA arrays and unknown repeat arrays (*URA*) (observed in chimpanzee and gorilla). *StSat* + *StSat*: association of interchromosomal *StSat* (observed in chimpanzee). *StSat* + *NRA*: association between StSat and non-repeat segments (*NRS*) (observed in chimpanzee). *INTRA*m: intercalary rDNA arrays (*INTRA*) in metacentric chromosome (m) (observed in Japanese and Rhesus macaques).

**Figure 2 cells-09-00971-f002:**
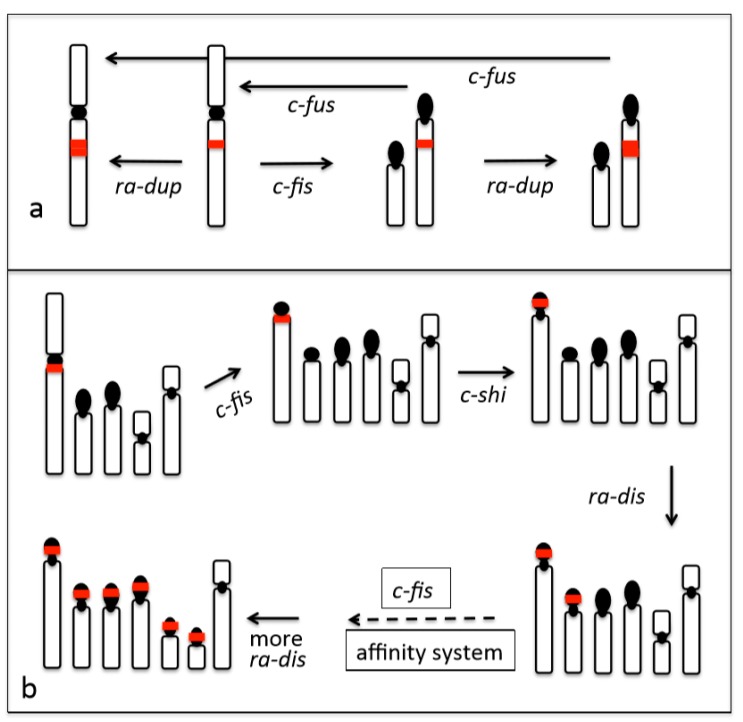
Plausible pathways of chromosome evolution and accompanied dispersion of rDNA. (**a**) rDNA located at an intercalary site in a chromosome arm. (**b**) rDNA located in a constitutive heterochromatin block close to the centromere. *c-fus*: centric fusion. *c-fis*: centric fission. *ra-dup*: duplication of rDNA array. *c-shi*: centromere shift. *ra-dis*: dispersion of rDNA array. affinity system: affinity among rDNA and other repeat arrays. Red blocks represent rDNA arrays. Black blocks represent constitutive heterochromatin. Black circles represent centromeres. Arrows indicate direction of chromosome change. See the text and [47] for the details.

**Figure 3 cells-09-00971-f003:**
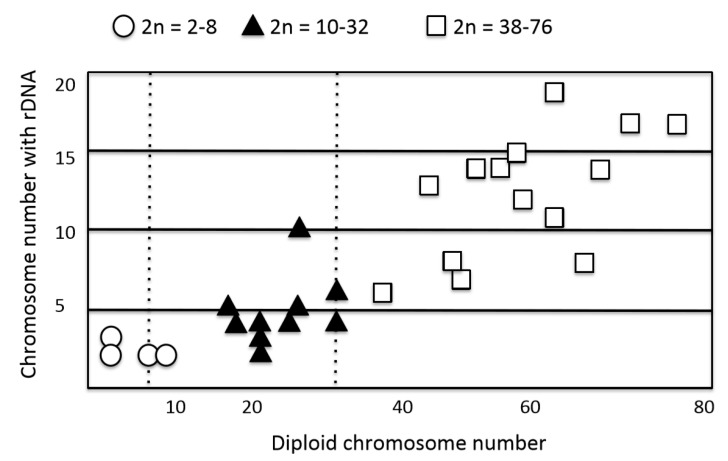
Relationship between diploid chromosome number and NOR chromosome number detected using rDNA probes in Australian bulldog ants. The *gulosa* group: *M. arnoldi* 2n = 53 − 66; *M. forficate* 2n = 52, 54; *M. gulosa* 2n = 38; *M. pavida* 2n = 44; *M. simillima* 2n = 70; *M. vindex* 2n = 74, 76. The *mandibularis* group: *M. fuvipes* 2n = 48, 50; *M. mandibularis* 2n = 56. The *pilosula* group: *M. banksi* 2n = 10; *M. chasei* 2n = 47; *M. croslandi* 2n = 3, 4; *M. haskinsorum* 2n = 18; *M. imaii* 2n = 8; *M. michaelseni* 2n = 27; *M. occidentalis* 2n = 64; *M. pilosula* 2n = 19, 22, 23, 32.

**Figure 4 cells-09-00971-f004:**
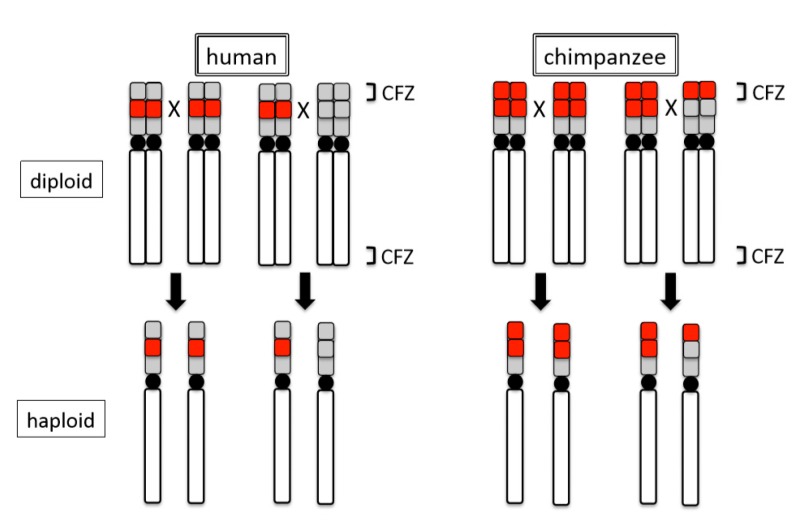
Difference in recombination at NOR loci between human and chimpanzee. Red blocks represent rDNA arrays. Grey blocks represent non-rDNA repeat arrays. CFZ: chiasma-free zone. ‘X’ indicates an assumed recombination site. Black arrows show possible recombinants. Black circles represent centromeres.

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
