# Peer review of "Chromosome Dynamics Regulating Genomic Dispersion and Alteration of Nucleolus Organizer Regions (NORs)"

_cells, 2020, doi:10.3390/cells9040971_

Round 1

Reviewer 1 Report

This review is on the posible constraints that seem to govern the chromosomal and genomic localization and dispertion dynamics of the NORs in the evolving genomes. It also includes an hypothesis on the matter.

The work can even be seen as in tune with the new tendency towards 3D genome analyses.

I only have a couple of suggestions:

1- in page 4 line 76 the sentence "...are required carefull…" should be changed to "...require carefull…"

2- in page 7 line 238: the statment about previous observations needs a reference.

All the best

Author Response

Reviewer 1:

This review is on the possible constraints that seem to govern the chromosomal and genomic localization and dispersion dynamics of the NORs in the evolving genomes. It also includes an hypothesis on the matter.

The work can even be seen as in tune with the new tendency towards 3D genome analyses.

I only have a couple of suggestions: (My direct replies and opinions are written in green, changed words and sentences are written in red, and new text is denoted by blue. Locations of revisions are indicated in brown.)

  • in page 4 line 76 the sentence "...are required carefull…" should be changed to "...require carefull…"

I changed “are required careful” to “require careful” according to your comment. See line 81 of the revised manuscript (MS) with revision history (RH).

  • in page 7 line 238: the statement about previous observations needs a reference.

Although I was not able to identify with certainty the sentence pointed out by you, if it is “7 times more frequently in chimpanzees than in humans”, a reference of No. 44 has been inserted after the sentence. See line 285 of the revised MS with RH.

Reviewer 2 Report

This review is about the evolutionary dynamics of nucleolus organizer regions (NORs). Using examples predominantly from primates and to a lesser extent from ants, the author intends to show that affinity between NORs and other repeat arrays during meiosis is the key factor triggering NOR genomic dispersion and evolution.

I fully agree that NORs deserve attention and are of interest with regards chromosomal rearrangements and evolution. The author has published on similar topics earlier and overall, the ideas presented in the manuscript have scientific merit. However, I have serious concerns about how these ideas are presented.

Major comments

  • The manuscript is supposed to be a review but is presented as a ‘hybrid’ between a review and a research paper. Unfortunately, it is lacking factual and scientific accuracy and thoroughness in both respects. For example, as a research paper, it lacks details for methodology and results. Also, the author writes “my colleagues and I ….” But gives no references and is not including the names of the colleagues as co-authors. On the other hand, as a review manuscript, it lacks information from contemporary studies of the past decade (except a few by the author). For example, out of 51 references, only two have been published after 2010, while the majority are from the 1990s or earlier.
  • The manuscript is too speculative and many claims by the author are not supported by references.
  • Several other observations or concepts are not provided with references. For example (but not limited to this), the sentence on page 3, lines 123-124 “Chromosomal configurations postulated from observations of molecular cytogenetic studies are depicted in Figure 1” must be supported by references.
  • The manuscript contains factual mistakes and lacks details in presented facts. For example (but not limited to these),
    • page 1, lines 35-37 is misleading and gives a wrong impression as if protein synthesis takes place in nucleolus;
    • It is true that NORs have argylophilic property (page 1, lines 40-41) but the author does not explain why it is so;
    • The author draws broad conclusions based on an extremely small set of species studied. The latter include mainly primates (known to have quite particular contents of repeat sequences), while the data on Bulldog ants is very limited. Furthermore, none of the species is named by their scientific name. Thus, it is not clear which Bulldog ants are in question. This is of particular importance because Myrmecia pilosula is known to have one of the lowest recorded diploid numbers among eukaryotes. All in all, for the kind of conclusions the author has made, a much broader collection of species should be included.
    • Figure 3 requires information about particular species and their diploid numbers.
    • There is no explanation why the author(s) used human probe for 18S rDNA and ant probe for 28S rDNA. Both genes are in the same tandemly repeated syntenic cluster. It may be worthwhile to briefly introduce to readers with the structure of the 18S-5.8S-28S locus.
    • The author discusses NORs together with various types of repeats. The latter are presented vaguely (“other arrays” “unknown repeat DNA”) and without proper reference. Sequence information for human, primates and many other species genomes has expanded and improved over the past decade and should be reflected in the manuscript. For example, for human/primate subterminal satellite repeats there are multiple contemporary publications, e.g., PMID 31273383
    • The author should explain the basis of the assumption that there is high recombination rate at terminal repeats and NOR associations. Recombination depends on double stranded breaks. Is there an evidence about higher rate of DS breaks at rDNA sites? Furthermore, recombination rates are different in different species, even between sexes of the same species. This needs to be properly addressed and with appropriate references.
    • Page 6, lines 202-203: the author’s claim that “more acrocentrics lead to more rDNA loci” needs critical revision. For example, just a quick check on three mammalian species with completely acrocentric autosomal genomes does not support this: mouse, 2n=40, six rDNA sites; cattle, 2n=60, five rDNA sites; dog, 2n=78, three autosomal and one Y rDNA.

Author Response

Reviewer 2:

This review is about the evolutionary dynamics of nucleolus organizer regions (NORs). Using examples predominantly from primates and to a lesser extent from ants, the author intends to show that affinity between NORs and other repeat arrays during meiosis is the key factor triggering NOR genomic dispersion and evolution.

I fully agree that NORs deserve attention and are of interest with regards chromosomal rearrangements and evolution. The author has published on similar topics earlier and overall, the ideas presented in the manuscript have scientific merit. However, I have serious concerns about how these ideas are presented.

(My direct replies and opinions are written in green, changed words and sentences are written in red, and new text is denoted by blue. Locations of revisions are indicated in brown.)

Major comments

  • 1) The manuscript is supposed to be a review but is presented as a ‘hybrid’ between a review and a research paper. Unfortunately, it is lacking factual and scientific accuracy and thoroughness in both respects. For example, as a research paper, it lacks details for methodology and results. Also, the author writes “my colleagues and I ….” But gives no references and is not including the names of the colleagues as co-authors. On the other hand, as a review manuscript, it lacks information from contemporary studies of the past decade (except a few by the author). For example, out of 51 references, only two have been published after 2010, while the majority are from the 1990s or earlier.

-> Your comments are very true. I myself also feel the somewhat intermediate nature of the manuscript. It stems from my intention to design it as a perspective report based on our own papers. (In particular, I aimed it as a perspective elated to chromosome dynamics of rDNA, or NORs, genomic dispersion.) However, the journal does not specifically offer a “Perspective” submission option. Following your recommendation, I have made revisions to make it more closely follow a “Review”. Namely, I have removed nearly all of the technical details of the methods, and I have added significant numbers of more recent references. I also included many papers written with my colleagues as references and listed the representative colleague names in acknowledgement section.

  • 2) The manuscript is too speculative and many claims by the author are not supported by references.

-> I developed my ideas mainly based on our own papers, as intended for the Perspective described above. Thus, the ideas and claims are indeed perspectives built directly upon our own observations. Without such extrapolations from our previous studies, the manuscript would not exist. However, your point is well taken. To buffer this situation, I have expanded some parts of the discussion and, where possible, added references to the manuscript.

  • 3) Several other observations or concepts are not provided with references. For example (but not limited to this), the sentence on page 3, lines 123-124 “Chromosomal configurations postulated from observations of molecular cytogenetic studies are depicted in Figure 1” must be supported by references.

-> Following your recommendation, I have added 8 references and added more discussion for the postulated chromosomal configurations. I have also added further discussion related to chromosomal configuration and genomic dispersion of rDNA. See lines 132-148 of the revised MS with RH.

  • The manuscript contains factual mistakes and lacks details in presented facts. For example (but not limited to these),
    • 4) page 1, lines 35-37 is misleading and gives a wrong impression as if protein synthesis takes place in nucleolus; -> I changed as follows: contain the tandem rDNA arrays responsible for composing the nucleolus, which is essential the site for all protein synthesis of synthesis and processing ribosomal proteins and rRNA [reviewed by 6]. See lines 36-37 of the revised MS with RH.
    • 5) It is true that NORs have argylophilic property (page 1, lines 40-41) but the author does not explain why it is so; -> I added an explanation and a reference as follows: an argyrophilic property of stainability to non-histone proteins in sulfhydryl and disulfide groups [e.g., 8, 9, 10 (Buys and Osinga 1980)]. See lines 41-43 of the revised MS with RH.
    • 6) The author draws broad conclusions based on an extremely small set of species studied. The latter include mainly primates (known to have quite particular contents of repeat sequences), while the data on Bulldog ants is very limited. Furthermore, none of the species is named by their scientific name. Thus, it is not clear which Bulldog ants are in question. This is of particular importance because Myrmecia pilosula is known to have one of the lowest recorded diploid numbers among eukaryotes. All in all, for the kind of conclusions the author has made, a much broader collection of species should be included.; -> I agree with your comments and thus added species names used for analysis of bulldog ants. In addition, I addedmore discussion in the appropriate regions as follows: the (genus Myrmecia). This ant species complex is very useful for investigating chromosome evolution, because they are species complex with similar morphology but a wide range of chromosome number (2n = 2-76) [48] (Imai et al. 1994). See lines 233-235 of revised MS with RH.
    • 7) Figure 3 requires information about particular species and their diploid numbers. Species names and chromosome numbers are added to the legend of Figure 3. See lines 255-259 of revised MS with RH.
    • 8) There is no explanation why the author(s) used human probe for 18S rDNA and ant probe for 28S rDNA. Both genes are in the same tandemly repeated syntenic cluster. It may be worthwhile to briefly introduce to readers with the structure of the 18S-5.8S-28S locus. -> I explained this in method section as follows: Eukaryotic rDNA consists of tandem repeat gene segments composed of 18S, 5.8S, and 28S tracts. To detect localization of NORs in mammals and ants, my colleagues and I therefore used probes of human 18S rDNA (kindly sipplied from Dr. AK Godwin) and ant 28S rDNA clones (M. closlandi (2n=2) rDNA (pMc.r2)), respectively, in FISH analysis [e.g., 19, 20]. See lines 66-69 of revised MS with RH.
    • 9) The author discusses NORs together with various types of repeats. The latter are presented vaguely (“other arrays” “unknown repeat DNA”) and without proper reference. Sequence information for human, primates and many other species genomes has expanded and improved over the past decade and should be reflected in the manuscript. For example, for human/primate subterminal satellite repeats there are multiple contemporary publications, e.g., PMID 31273383 -> I included the recommended paper as a reference to add new information for suggesting unknown repeat arrays and described the information in the text as follows: Although unknown repeat arrays (URAs) observed as heterochromatic blocks were not identified at the sequence level in our previous studies, some satellite repeats with motifs like (AATGG)n, (ACTCC)n, and (AAAG)n are suggested as candidates based on recent genome sequence analysis [37] (e.g., Cechova et al. 2019 and references therein). See lines 192-195 of revised MS with RH.
    • 10) The author should explain the basis of the assumption that there is high recombination rate at terminal repeats and NOR associations. Recombination depends on double stranded breaks. Is there an evidence about higher rate of DS breaks at rDNA sites? Furthermore, recombination rates are different in different species, even between sexes of the same species. This needs to be properly addressed and with appropriate references. -> I added information regarding the relationship between variations of rDNA localization and double-strand breaks (DSBs) and described it as follows: The rate of ectopic recombination is much less than that of homologous recombination, as inferred from successive observations indicating that heterozygotic variations of rDNA localization are sporadic, as mentioned above. Indeed, double-strand breaks (DSBs) at the sites of repetitive sequences can be a potent source of genomic alteration [35, 36] (Vader et al. 2012, van Sluis and McStay 2015). The local chromatin environment very likely plays a role in promorting repair of DSBs within rDNA via he homologous recombination pathway [36] (van Sluis and McStay 2015). See lines 176-181 of revised MS with RH.

11) Page 6, lines 202-203: the author’s claim that “more acrocentrics lead to more rDNA loci” needs critical revision. For example, just a quick check on three mammalian species with completely acrocentric autosomal genomes does not support this: mouse, 2n=40, six rDNA sites; cattle, 2n=60, five rDNA sites; dog, 2n=78, three autosomal and one Y rDNA. -> I added to the discussion to more completely and specifically explain my position as follows: The reason why is that the number ofrDNA-chromosome is 2 in the “primitive” karyotypes 2n=2-8, but has increased gradually from 2 to 19 in karyotypes with 2n=10-32 and 2n=38-76. Figure 3 depicts the increase of diploid number (2 to 76), – via increasing number of acrocentrics by centric fission – as correlated with change of rDNA-chromosome number (2 to 19) [see also 47 (Hirai et al. 1996)]. If the heterochromatin arrays of the acrocentric short arms did not have affinity with the rDNA arrays, however, the rDNA arrays would not have been able to move to those short arms via association and ectopic recombination. See lines 243-249 of revised MS with RH.

Reviewer 3 Report

This review describes the possible mechanisms implicated in NORs genomic distribution. The review is of interest but not easy to follow. It also mainly focuses on observations made from cell biology approaches in primates, but also mention NOR distribution in Australian bulldog ants.

First, I think that for a review article, the method section is very detailed and not necessary. The review would benefit from focusing on the concepts discussed. It also took me some time to understand what the author mean by affinity and non-affinity… a better definition at the beginning of the review should to make it more clear.

In general, the review uses very descriptive data and do not cite the potential molecular mechanisms that could be implicated for example in rDNA genomic dispersion for example… just mentioning “using different molecular effect systems”. This is too vague!

The review would also benefit from data obtained in other mammal or insect species to better strength the author’s hypothesis.

For the figure 1 and 2, presence of the acronym definition will also help the reader to understand them more quickly.  Also, figure 1 is a mix of the different organization that can be found in primates. In that context, I would also include a summary table explaining the different possibilities observed in the different primate species.

Is there any data on the localization of the various NOR and repeats at interphase nuclei to better correlate the physical association of these various repeats in the cell? Do they occupy different nuclear compartments? For example, are all NORs associating with nucleoli? Are repeated sequences like StSat more associating the nuclear periphery for example? NOR are not all expressed in the most cells. In that case, NOR evolution is certainly linked to their transcriptional activity. The author do not mention this link but since active NOR associate with the nucleoli, while inactive NOR do not necessary associate with the nucleoli. In consequence, they usually occupy different nuclear compartments. The author should discuss the potential influence of this transcriptional activity on NOR evolution.

The idea that repeat arrays can act as natural constraints to the genomic dispersion is interesting… But is there any idea about the various influence of the different repeats in the expression of the neighboring genes? Can they differentially influence the local chromatin states?

What makes the author saying that physical association of rDNA is likely the engine of concerted evolution? Are they studies showing more homogenous NOR composition in the context of RA+RA for example?

In the Australian bulldog ants, since the genomic context flanking NORs is similar (centromeric sequences), could homologous recombination be another mechanism explaining the NOR dispersion and/or duplication?

The author also mention the intra-specific variation of NOR distribution. Is there any data about their transgenerational inheritance?

The author presents the transposition as the only system leading to rDNA dispersion in plants. Is the author sure about this fact?

Author Response

This review describes the possible mechanisms implicated in NORs genomic distribution. The review is of interest but not easy to follow. It also mainly focuses on observations made from cell biology approaches in primates, but also mention NOR distribution in Australian bulldog ants.

 (My direct replies and opinions are written in green, changed words and sentences are written in red, and new text is denoted by blue. Locations of revisions are indicated in brown.)

1) First, I think that for a review article, the method section is very detailed and not necessary. The review would benefit from focusing on the concepts discussed. It also took me some time to understand what the author mean by affinity and non-affinity… a better definition at the beginning of the review should to make it more clear. -> I defined ‘affinity’ and ‘non-affinity’ at their first appearance with a following sentence, and I deleted the details of method: “Affinity” used here is defined as attractive force, or junction power, among DNA segments. See lines 63-64, 72-78 (deleted), 85-95 (deleted), and 99-105 of revised MS with RH.

2) In general, the review uses very descriptive data and do not cite the potential molecular mechanisms that could be implicated for example in rDNA genomic dispersion for example… just mentioning “using different molecular effect systems”. This is too vague! -> the sentence “different molecular effect systems” was replaced with “genome composition”. See line 290 of revised MS with RH.

3) The review would also benefit from data obtained in other mammal or insect species to better strength the author’s hypothesis. -> I added 11 references dealing with rDNA dispersion in plants, insects, and mammals. See lines 324-330 of revised MS with RH.

4) For the figure 1 and 2, presence of the acronym definition will also help the reader to understand them more quickly.  Also, figure 1 is a mix of the different organization that can be found in primates. In that context, I would also include a summary table explaining the different possibilities observed in the different primate species. -> I agree with your opinions. I described species names and chromosome numbers, and I provided more recommended information in the Figure 1 legend. See Figure 1.

5) Is there any data on the localization of the various NOR and repeats at interphase nuclei to better correlate the physical association of these various repeats in the cell? Do they occupy different nuclear compartments? For example, are all NORs associating with nucleoli? Are repeated sequences like StSat more associating the nuclear periphery for example? NOR are not all expressed in the most cells. In that case, NOR evolution is certainly linked to their transcriptional activity. The author do not mention this link but since active NOR associate with the nucleoli, while inactive NOR do not necessary associate with the nucleoli. In consequence, they usually occupy different nuclear compartments. The author should discuss the potential influence of this transcriptional activity on NOR evolution. -> I think your comments are very important, but unfortunately since I did not concentrate on that point, I have not observed the relationship of nucleoli and NOR variability in each chromosome. Though I cannot provide a detailed discussion, I can provide a brief description of the relationship between nucleolus and rDNA loci in bulldog ants as follows: Physical contacts between chromatins are related to their tropism by the affinity/non-affinity system, which stems from DNA traits. This ultimately brings about genomic reaction, including dispersion and/or recombination, and generates genetic divergence. Structural patterns also likely represent important indexes reflecting the relationship between transcriptional activity and NOR evolution. The most direct connection is that between NORs and nucleolus. In cytological observations of Australian bulldog ants the relationship between nucleolus formation and rDNA loci revealed that the nucleolus (visualized by silver nitrate staining) appeared only in the vicinity of an rDNA locus in interphase to prophase. That is, only an rDNA array associates with the active nucleolus in the formation of nucleolus. This phenomenon was observed in all species examined [20]. The localization of NORs in the nucleolus, and its associations with the nucleolus, are closely linked to the transcription activity of rDNA [76]. Indeed, future chromosomal studies related to rDNA and NORs need to pay close attention to cellular states and epigenetic states of the genome that are driven by the rDNA/nucleolus [77]. These sentences are inserted into the final paragraph of conclusion section.

6) The idea that repeat arrays can act as natural constraints to the genomic dispersion is interesting… But is there any idea about the various influence of the different repeats in the expression of the neighboring genes? Can they differentially influence the local chromatin states? -> Since I have never considered the expression of the genes neighboring repetitive segments, I cannot discuss the influence of the position effect on genes. However, segments of repeat sequences (e.g., StSat) might influence homologous and heterologous recombination in the regions nearby the blocks of repetitive sequences.

7) What makes the author saying that physical association of rDNA is likely the engine of concerted evolution? Are they studies showing more homogenous NOR composition in the context of RA+RA for example? -> As I have not investigated NORs at the sequence level, the degree of homogenization of each NOR locus is not clear yet. However, our previous study indicated that chimpanzees and bonobos have maintained sequence similarity of StSat, a repeat array which shares some characteristics with rDNA. Therefore, they probably have maintained the sequence by a homogenization system even after speciation.

8) In the Australian bulldog ants, since the genomic context flanking NORs is similar (centromeric sequences), could homologous recombination be another mechanism explaining the NOR dispersion and/or duplication? -> Reciprocal translocation is also suggested for inferring NOR dispersion. However, I did not argue it in this review because reciprocal translocations are less likely when chromosome number increases. Certainly, several pathways for chromosome evolution should be considered but in the case of the species complex of bulldog ants, the increasing pathway of chromosome number seems to be mostly relevant to considerations of chromosome evolution.

9) The author also mention the intra-specific variation of NOR distribution. Is there any data about their transgenerational inheritance? -> I have observed inheritance of NOR distribution (actually deletion of a rDNA locus) in a human family linage.

10) The author presents the transposition as the only system leading to rDNA dispersion in plants. Is the author sure about this fact? -> I added some information and sentences related to the revised discussion of changes of rDNA sites in plants as follows: Indeed, transposition with mobile elements, rather than structural rearrangement, is known to be a decisive mechanism for rDNA genomic dispersion in plants [e.g., 64-68], insects [e.g., 69-71] and mammals [e.g., 63] – and it might occasionally operate as a mechanism of “molecular effect” in organisms. Certainly, changes of the rDNA sites do not vary randomly and seem to be mediated mainly by the bouquet configuration because the sites are preferentially localized at the terminal regions (73.8%) [72]. Yet, since the mammals surveyed to date do not exhibit direct association between transposable elements and rDNA arrays [73], it is most likely that the affinity/non-affinity system described here between rDNA and other repeat arrays is driving related cytogenetic changes. See lines 324-330 of revised MS with RH.

Round 2

Reviewer 2 Report

The authors have adequately responded to comments and revised the manuscript